# Thrombotic Adverse Events Reported for Moderna, Pfizer and Oxford-AstraZeneca COVID-19 Vaccines: Comparison of Occurrence and Clinical Outcomes in the EudraVigilance Database

**DOI:** 10.3390/vaccines9111326

**Published:** 2021-11-15

**Authors:** Mansour Tobaiqy, Katie MacLure, Hajer Elkout, Derek Stewart

**Affiliations:** 1Department of Pharmacology, College of Medicine, University of Jeddah, Jeddah P.O. Box 45311, Saudi Arabia; 2Independent Researcher, Aberdeen AB32 6RU, UK; katiemaclure@outlook.com; 3Department of Family and Community Medicine, Medical Faculty, University of Tripoli, Tripoli P.O. Box 13662, Libya; h.elkout@uot.edu.ly; 4College of Pharmacy, QU Health, Qatar University, Doha P.O. Box 2713, Qatar; d.stewart@qu.edu.qa

**Keywords:** adverse events, adverse drug reaction, thrombotic, COVID-19 vaccine, Pfizer, Moderna, AstraZenaca

## Abstract

Vaccination against COVID-19 is the cornerstone of controlling and mitigating the ongoing pandemic. Thrombotic adverse events linked to Moderna, Pfizer and the Oxford-AstraZeneca vaccine have been documented and described as extremely rare. While the Oxford-AstraZeneca vaccine has received much of the attention, the other vaccines should not go unchallenged. This study aimed to determine the frequency of reported thrombotic adverse events and clinical outcomes for these three COVID-19 vaccines, namely, Moderna, Pfizer and Oxford-AstraZeneca. A retrospective descriptive analysis was conducted of spontaneous reports for Moderna, Pfizer and Oxford-AstraZeneca COVID-19 vaccines submitted to the EudraVigilance database in the period from 17 February to 14 June 2021. There were 729,496 adverse events for the three vaccines, of which 3420 were thrombotic, mainly Oxford-AstraZeneca (n = 1988; 58.1%) followed by Pfizer (n = 1096; 32.0%) and Moderna (n = 336; 9.8%). As serious adverse events, there were 705 reports of pulmonary embolism for the three vaccines, of which 130 reports (18.4%) were for Moderna, 226 reports (32.1%) for Pfizer and 349 (49.5%) for Oxford-AstraZeneca vaccines. The occurrence of pulmonary embolism is significantly associated with a fatal outcome (*p* ≤ 0.001). Sixty-three fatalities were recorded (n = 63/3420; 1.8%), of which Moderna (n = 6), Pfizer (n = 25) and Oxford-AstraZeneca (n = 32).

## 1. Introduction

Vaccination against Coronavirus Disease 2019 (COVID-19) is the cornerstone of controlling and mitigating the ongoing pandemic [1]. Thrombotic adverse events include venous thrombosis, thrombocytopenia and ischemic stroke, which have been reported in patients in the days following their first dose of vaccine with varying consequences [2,3,4,5]. Thrombocytopenia has been reported as being associated with the Oxford-AstraZeneca vaccine [4]. This is a clinical syndrome which manifests as a high level of antibodies against platelet factor 4 (PF4) unrelated to heparin-induced thrombocytopenia (HIT). It is referred to as vaccine-induced immune thrombotic thrombocytopenia (VITT). Similar to HIT, bleeding is rare with both HIT and VITT; however, both are associated with thromboembolic complications involving the arterial and venous systems [4,6]. A rare arterial thrombosis may also be considered an adverse event associated with COVID-19 vaccination as reported in a recent case of malignant cerebral infarct that has been associated with thrombocytopenia [7]. High serum levels of antibodies to PF4-polyanion complexes have been observed in two young, healthy, adult women within 10 days of receiving Oxford-AstraZeneca vaccination [7].

While the Oxford-AstraZeneca vaccine has been the focus, particularly for blood clotting events, other vaccines have also been reported for thrombotic adverse events. This has included Johnson & Johnson’s Janssen (J&J/Janssen) COVID-19 vaccine, which was suspended by the Food and Drug Administration (FDA) in the United States (US) and then resumed when the FDA concluded that its’ benefits outweighed the known and potential risks [5,8]. A recent pharmacovigilance study reported 28 potential thrombotic adverse events out of 54,571 patient vaccinations linked to the Oxford-AstraZeneca vaccine giving an occurrence rate of 0.05% [9]. The study was also based on data from the EudraVigilance (EV) database in the period from 17 February to 12 March 2021 [9]. Among these, three fatalities were related to pulmonary embolism with one fatality to thrombosis. The same study identified 29 deep venous thromboses (DVTs) associated with the Moderna vaccine, of which 12 also had pulmonary embolism; there were no reported deaths or fatal events but there were 13 DVTs with the Pfizer vaccine in the same study period [9]. 

There have been several articles which looked at sex and age differences of people who had experienced adverse events [10,11,12,13,14,15,16]. Some suggest women have stronger immune systems [10]. In one article comparing females and males, the literature described that females compared to males have greater inflammatory, antiviral, and humoral immune responses [11]. In females, oestrogen is a potential ally in alleviating severe acute respiratory syndrome coronavirus 2 (SARS-CoV-2) disease [11]. In males, testosterone reduces vaccination response and depresses the cytokine response [11]. That same article also covers older patients, finding that “older patients, and in particular, in female older patients, it has been reported a progressive functional decline in the immune systems” [11]. In some countries, differences in gender roles find women with “low autonomy, labour responsibilities, and unpaid care burdens were reasons for gendered barriers to vaccination that disadvantaged women” [12]. Others suggest hormone levels in both men and women impact vaccine tolerance [13]. Men have been found to experience greater severity of disease and higher mortality [13,15]. However, this set of articles did not consider differences between vaccines nor the confounding impact of co-morbidities [10,11,12,13,14,15,16].

Reports comparing the occurrence and clinical outcomes of thrombotic adverse events with these three vaccines remain scarce. Adverse events related to vaccines are the most frequently reported among the collected spontaneous reports logged with the EV database [17]. As COVID-19 vaccination is broadened to younger age groups, and top-up vaccination for older age groups, decision makers need current evidence on which to base vaccine recommendations. 

Therefore, the aim of this study was to compare the reported occurrence of thrombotic adverse events and clinical outcomes for three COVID-19 vaccines, namely Moderna, Pfizer and Oxford-AstraZeneca, based on reports from the EudraVigilance database with a particular focus on pulmonary embolism.

## 2. Materials and Methods

A retrospective descriptive analysis was conducted of spontaneous reports for Moderna, Pfizer and Oxford-AstraZeneca COVID-19 vaccines submitted to the EudraVigilance database (https://www.ema.europa.eu/en/human-regulatory/research-development/pharmacovigilance/eudravigilance, accessed on 23 June 2021) for the period 17 February to 14 June 2021.

### 2.1. Data Sources and Setting 

Data were extracted from EV, a European Economic Area (EEA) wide open access pharmacovigilance database, which collates adverse event reports from non-healthcare and healthcare professionals in more than 50 countries. The line listing section of spontaneous reports submitted to the EV for COVID-19 vaccines Moderna, Pfizer (Tozinameran) and Oxford-AstraZeneca (ChAdOx1-S) was accessed. The following search terms were applied: thrombosis; embolism; thromboembolism; embolic; and thrombotic. Taking pulmonary embolism as a focus, serious adverse events were identified and further analysed according to age, sex and type of vaccine. The reports identified were then exported to Microsoft Excel files and classified according to the following variables: age (age groups: ≥85 years old, 65–84 and 18–64 years old), sex, month of reporting, origin of the report, reporter’s profession (healthcare professional or non-healthcare professional) and concomitant conditions. Data were tabulated and presented along with the clinical outcomes which were categorised into four sections: (i) recovered, (ii) recovering, (iii) not yet recovered, and (iv) fatal outcome. 

Ethical review was not required for the open access data. The access policy of European Medicines Agency (EMA) states that “No authorisation for accessing the ICSR (Level 1) data set by means of the adrreports.eu portal is required, i.e., all academic researchers can access adverse reaction data of interest” [17].

### 2.2. Statistical Analysis

Data were analysed using descriptive statistics to determine the study population characteristics. Variables were reported as absolute numbers and percentages. All analyses were performed using SPSS for Windows 20.0 (SPSS, Chicago, IL, USA). Differences in proportions between the groups were compared with the Chi-square test and monthly trends in reporting were analysed. A *p*-value of less than 0.05 was considered statistically significant. 

## 3. Results

### 3.1. Frequency of Adverse Events

During the study period, a total of 729,496 adverse events were reported for the three vaccines: Oxford-AstraZeneca (n = 337,712; 46.3%), Pfizer (n = 311,364; 42.7%) and Moderna (n = 80,420; 11.0%). 

### 3.2. Frequency of Thrombotic Adverse Events Associated with the Three Vaccines

As shown in Table 1, of those reports, there were 3420 thromboembolic adverse events reported, with 1988 (58.1%) for Oxford-AstraZeneca, 1096 (32%) for Pfizer and 336 (9.8%) for Moderna. Table 1 gives the general characteristics of the reports and Table 2 the number of monthly reports including reports with a fatal outcome.

### 3.3. Pulmonary Embolism Reports

There were 705 reports for the three vaccines that included pulmonary embolism (Table 3). Occurrence of pulmonary embolism is significantly associated with a fatal outcome (*p* ≤ 0.001). 

### 3.4. Outcome of the Thrombotic Adverse Events 

Of the total 705 reports, 63 reports were for a fatal outcome; of those, six cases (1.8%) were for the Moderna vaccine, 25 (2.3%) for Pfizer vaccine and 32 reports (1.6%) for the Oxford-AstraZeneca vaccine. Out of the total 63 reports of fatal outcomes, 38 (60.3%) had pulmonary embolism compared to 25 (39.7%) fatal reports for other thrombolytic adverse events. Within the type of vaccine, of the 38 reports with fatal outcome and pulmonary embolism, three of six reports were recorded for Moderna, 17 out of 25 for Pfizer and 18 out of 32 reports for Oxford-AstraZeneca vaccines, comprising 50.0%, 68.0% and 56.3% of the total reports with fatal outcome for each vaccine, respectively.

Table 3 presents the number and proportions of the range of outcomes among the three COVID-19 vaccines. Moreover, the analyses also showed 589 reports with an unknown outcome, with Moderna (n = 96; 28.6%), Pfizer (n = 146; 13.3%), and for Oxford-AstraZeneca (n = 347; 17.5%) vaccines; 369 reports for recovered patients for Moderna (n = 50; 14.9%), for Pfizer (n = 125; 11.4%) and for Oxford-AstraZeneca (n = 194; 9.8%) vaccines; 1279 reports for recovering patients with Moderna (n = 40; 11.9%), for Pfizer (n = 449; 71%), and for Oxford-AstraZeneca (n = 790; 39.7%) vaccines. 1120 reports were defined as having a not recovered outcome with Moderna (n = 144; 42.9%), for Pfizer (n = 351; 32%) and for Oxford-AstraZeneca (n = 625; 31.4%) vaccines. 

Regarding age, those in the 65–84 years age group reported higher fatal outcomes than other age groups (55.5% of all deaths). When examined for each type of vaccine, about half of the reports with a fatal outcome occurred in the same age group. However, only the Moderna vaccine showed significant association between age and a fatal outcome (*p* value = 0·008) (Table 3 and Table 4). In addition, the reports contained other concomitant adverse events such as thrombocytopenia, arrhythmia, and cerebro-vascular thrombosis (Table 5); however, data on concomitant drugs was unavailable.

## 4. Discussion

The development of safe, effective, affordable vaccines against COVID-19 remains the cornerstone to mitigating this pandemic. By 20 August 2021, more than 4.89 billion doses of different COVID-19 vaccines have been administered across the globe, and over 40 candidate vaccines were in human trials [18,19]. Yet there is variable hesitancy, fear and anxiety [20] due to perceived risk of vaccination [21]. A recent UK study found increased risks of both haematological and vascular events that led to hospital admission or death following first doses of both Pfizer and Oxford-AstraZeneca vaccines but also noted the risk was higher following COVID-19 infection [22]. 

This study identified 729,496 adverse events for the three vaccines under investigation (Moderna, Pfizer and Oxford-AstraZeneca) for the period of study from 17 February to 14 June 2021, of which 3420 were thrombotic adverse events: 336 (0.41%) for Moderna, 1096 (0.35%) Pfizer and 1988 (0.58%) for Oxford-AstraZeneca vaccines. In this study, thrombocytopenia with concomitant clinical conditions were reported 157 times with the three vaccines and was associated more with Oxford-AstraZeneca (n = 134, 85.3%) than with Pfizer (n = 17, 10.8%) and Moderna (n = 6, 3.8%). This finding has been demonstrated recently in a study that reported thrombotic events in 11 patients post-vaccination with Oxford-AstraZeneca and showed evidence of immune thrombotic thrombocytopenia mediated by a high level of platelet-activating antibodies against PF4 [4].

In this study, the majority (53.8%) of potential thrombotic adverse events that were linked to the Oxford-AstraZeneca vaccine were reported in the age group 18–64 years, whereas the majority of these events were reported in the age group of 65–84 years for Moderna (48.2%) and Pfizer vaccines (47.3%), respectively. In contrast, other research that used the EV database showed that reported thrombotic events were more prevalent in the age group <65 among Oxford-AstraZeneca vaccine recipients than Moderna or Pfizer vaccine recipients [23]. As a study reporting real world data from the EV database, this current study had no control over sample size so was not specifically powered to report on age.

Of note, there was a slight increase, not statistically significant, in the number of reports of female sex for Oxford-AstraZenaca (50.6%) and Pfizer (53.2%) vaccines. However, the Moderna vaccine had more reports linked to males (*p*-value 0.008). We have reported earlier approximately double the occurrence of potential thrombotic events reported in females (n = 19) than males (n = 9) for the Oxford-AstraZenaca vaccine [9]. Similarly, the clinical and laboratory data of 11 thrombotic patients post vaccination with Oxford-AstraZeneca were nine female patients with a median age of 36 years (range 22 to 49) [24]. Other articles have reported females experiencing more adverse events and with greater severity but did not differentiate between vaccines [10,14]. They did, however, emphasise the need for sex differences to be considered in vaccine programmes [10,14].

Most potential thrombotic adverse events were reported for Oxford-AstraZeneca (n = 1988; 58.1%). However, the present study and other pharmacovigilance research demonstrated that thrombosis with thrombocytopenia appears to occur with all three vaccines, with higher rates in those who had received the Oxford-AstraZeneca vaccine [23,25]. It has been suggested that administration of the Oxford-AstraZeneca vaccine should be considered carefully and only when the potential benefit outweighs any potential risks, particularly in patients with a history of cerebral venous sinus thrombosis, acquired or hereditary thrombophilia, or heparin-induced thrombocytopenia [26,27].

Pulmonary embolism (PE) is an acute complication of DVT and is considered a serious adverse event [28] and the third most common cause of cardiovascular death [29]. In this study, among the severe clinical outcomes associated with the three vaccines, 705 reports included pulmonary embolism with 63 fatalities; however, this finding should be interpreted with great caution given the nature of the present data and the passive reporting system used.

The limitations of this study, as with spontaneous reporting schemes of adverse reactions of both drugs and vaccines, is hampered by underreporting, over reporting and reporting bias. This makes it difficult to identify the true incidence of these events and the presence of multiple confounders which may not enable the assessment of the causality with higher specificity [30]. In addition, due to the nature of the database, it was not possible to know the denominator (the total number of vaccinated individuals for each type of vaccine), which hinders the analysis of the likelihood of the true occurrence of thrombotic adverse events for each vaccine. We were also unable to report possible concomitant drugs. Despite these limitations, the study used global real-world data and collected valuable information about three widely used vaccines, where more than two thirds of all reports were received from healthcare professionals, which increases the credibility and quality of reports.

People who are vaccine hesitant and reluctant to take any of the mentioned vaccines [31] should know that most COVID-19 vaccines are effective at preventing symptomatic infection including hospital admissions and severe disease [32,33]. The risk of COVID-19 vaccine related thrombotic events are minimal and likely manageable with available treatments [9]. Thrombotic adverse events reported for the three vaccines remains extremely rare [9,34]. 

## 5. Conclusions

In summary, thrombotic adverse events reported for the three vaccines remains extremely rare with multiple causative factors reported elsewhere as precipitating these events. Practicing vigilance and proper clinical management for the affected vaccines, as well as continuing to report adverse events, is essential. Future research should consider sex and age differences of adverse events with COVID-19 vaccines. Results can influence vaccination programmes and policy makers’ decision making.

## Figures and Tables

**Table 1 vaccines-09-01326-t001:** Summary of demographics of EudraVigilance database thrombotic adverse event reports (N = 3420) for Moderna, Pfizer and Oxford-AstraZeneca vaccines.

Variable		n (%)		
		Moderna 336 (9.8)	Pfizer 1096 (32)	Oxford-AstraZeneca 1988 (58.1)
Sex	Male	171 (50.9)	494 (45.1)	952 (47.9)
	Female	164 (48.8)	583 (53.2)	1006 (50.6)
	Not specified	1 (0.3)	19 (1.7)	30 (1.5)
Reporter profession	Healthcare			
	professional	293 (87.2)	804 (73.4)	1422 (71.5)
	Non-healthcare			
	professional	43 (12.8)	292 (26.6)	566 (28.5)
Geographical area	EU	96 (28.6)	768 (70.1)	877 (44.1)
	Non-EU/EEA *	240 (71.4)	328 (29.9)	1111 (55.9)
Age group of patients	18–64 years	140 (41.7)	374 (34.1)	1069 (53.8)
	65–84 years	162 (48.2)	519 (47.3)	725 (36.4)
	85 years and older	27 (8.0)	147 (13.4)	81 (4.1)
	Missing data	7 (2.1)	56 (5.1)	111 (5.6)

* EEA—European Economic Area.

**Table 2 vaccines-09-01326-t002:** Monthly distribution of thromboembolic EV reports for the three vaccines from 17 February to 14 June 2021.

Month	Number of Reports n			Reports with Fatal Outcome n (%)		
	Moderna	Pfizer	Oxford-AstraZeneca	Moderna	Pfizer	Oxford-AstraZeneca
February	6	24	0	0	0	0
March	29	127	319	0	0	4 (1.3)
April	47	285	591	2 (4.3)	7 (2.5)	2 (0.3)
May	169	481	758	4 (2.4)	8 (1.7)	15 (2.0)
June	85	173	319	0	10 (5.8)	10 (3.1)

**Table 3 vaccines-09-01326-t003:** Clinical outcome of all adverse events from 17 February to 14 June 2021 with demographic variables related to Moderna, Pfizer and Oxford-AstraZeneca vaccines with a focus on pulmonary embolism.

	Clinical Outcome of All Adverse Event Reports n (%)					Pulmonary Embolism Only n (%)
	Outcome Unknown	Recovered	Recovering	Not Yet Recovered	Fatal	
Type of vaccine						
Moderna (N = 336)	96 (28.6)	50 (14.9)	40 (11.9)	144 (42.9)	6 (1.8)	130 (38.7)
Pfizer (N = 1096)	146 (13.3)	125 (11.4)	449 (71.0)	351 (32.0)	25 (2.3)	226 (20.6)
Oxford-AstraZeneca (N = 1988)	347 (17.5)	194 (9.8)	790 (39.7)	625 (31.4)	32 (1.6)	349 (17.6)
Age group of patients						
18–64 years (N = 1583)	261 (16.5)	167 (10.5)	573 (36.2)	566 (35.8)	16 (1.0)	284 (17.9)
65–84 years (N = 1406)	238 (16.9)	153 (10.9)	559 (39.8)	421.29.9)	35 (2.5)	343 (24.4)
85 years and older (N = 255)	48 (18.8)	30 (11.9)	82 (32.2)	87 (34.1)	8 (3.1)	56 (22.0)
Not specified (N = 174)	42 (24.1)	18 (10.3)	65 (37.4)	45 (1.3)	4 (2.3)	22 (12.6)
Sex						
Male (N = 1617)	268 (16.6)	181 (11.2)	637 (39.4)	507 (31.4)	24 (1.5)	373 (23.1)
Female (N = 1753)	313 (17.9)	183 (10.4)	615 (35.1)	603 (34.4)	39 (2.2)	323 (18.4)
Not specified (N = 50)	8 (16.0)	4 (8.0)	28 (56.0)	10 (20.0)	··	··

**Table 4 vaccines-09-01326-t004:** Summary of reports with fatal outcome in relation to age and sex for each type of vaccine.

Type of Vaccine Fatal Outcome	Sex n (%)		Age n (%)			
	Male	Female	Not Specified	18–64	65–84	85 and Over
Moderna	1 (1.6)	5 (3.0)	..	..	4 (2.5)	2 (7.4)
Pfizer	10 (2.0)	15 (2.6)	..	5 (1.3)	17 (3.3)	3 (2.0)
Oxford-AstraZeneca	13 (1.4)	19 (1.9)	4 (3.6)	11 (1.0)	14 (1.9)	3 (3.7)

**Table 5 vaccines-09-01326-t005:** Concomitant clinical conditions and other thrombotic adverse events.

Concomitant	Moderna	Pfizer	Oxford-AstraZeneca	Total
Thrombocytopenia	6 (1.8)	18 (1.6)	136 (6.8)	157
Arrhythmia	7 (2.1)	5 (0.5)	8 (0.4)	29
Cerebro-vascular thrombosis	4 (1.2)	9 (0.8)	12 (0.6)	20

## Data Availability

Dataset openly available in the public domain.

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
