# Peer review of "Thrombotic Adverse Events Reported for Moderna, Pfizer and Oxford-AstraZeneca COVID-19 Vaccines: Comparison of Occurrence and Clinical Outcomes in the EudraVigilance Database"

_vaccines, 2021, doi:10.3390/vaccines9111326_

Round 1

Reviewer 1 Report

The manuscript titled “Thrombotic adverse events reported for Moderna, Pfizer and Oxford-AstraZeneca COVID-19 vaccines: comparison of occurrence and clinical outcomes in the EudraVigilance database” by Dr. Tobaiqy and colleagues aimed to determine the frequency of reported thrombotic adverse events and clinical outcomes for Moderna, Pfizer and Oxford-AstraZeneca COVID-19 vaccines. To this purpose, a retrospective descriptive analysis has been performed on vaccines submitted 85 to the EudraVigilance database for the period 17 February to 14 June 2021. The study identified 729, 496 adverse events for Moderna, Pfizer and Oxford-AstraZeneca, of which 3,420 thrombotic adverse events: 336 (0·41%) for Moderna, 1096 173 (0·35%) Pfizer and 1988 (0·58%) for Oxford-AstraZeneca vaccines.

In my opinion, this is an interesting work, well written and organized. This work is relevant to the field, as it increases our knowledge on adverse events occurring during COVID-19 vaccination. I have a few observations only and I am therefore recommending a minor revision.

General comment
1.    The presence of co-morbidities as risk factor for adverse events occurring during vaccination should be included and described in the introductive section, following lines 59-72
2.    The total number of vaccinated patients/individuals should be included throughout the text. In addition, adverse events rates should also be as % of the the total number of vaccinated patients/individuals, as described in lines 51-53 for the Oxford-AstraZeneca investigation.

Minor
Lines 31 and 63 COVID-19 and SARS-CoV-2should be Coronavirus disease 2019 (COVID-19) and severe acute respiratory syndrome coronavirus 2 (SARS-CoV-2), respectively, when mentioned for the first time. 
Line 34 Detailed information on the efficacy and adverse events of Moderna, Pfizer and Oxford-AstraZeneca COVID-19 vaccines is reported here (PMID:34578269). This supporting reference should be included 
Line 55. This sentence is lacking in supporting reference. This reference should be included PMID: 32490931
Lines 81-82 the countries covered by EudraVigilance database should be mentioned. At least in the method section
Line 86 I suggest including the official website. It would be helpful for the reader. 
Lines 86-87 the last sentence should be separated from the following subhead title
Line123 the number of total cases should be included in the table 1

Author Response

Many thanks for your valuable comments and suggestions

Reviewer 2 Report

Introduction

The presence of antibodies is just laboratory findings, but not manifestation of any syndrome. Please rephrase the sentence at page 1 line 41-42 "This condition manifests as high serum levels of antibodies to.."

Results

Table 1. Please modify the table-layout.

The first horizontal line (Moderna, Pfizer and Astra) must be in same layer.

Table 2. Please specify if the data presented as number (%). Moreover, For AstraZeneca, the data for June is presented only as %. Please correct.

Page 4, line 128

Please clarify the sentence "There were 705 reports of pulmonary embolism during the study period for the three vaccines that included deep vein thrombosis with and without pulmonary embolism".

Line 131-132.

The data about the number and percent of pulmonary embolism incidents in the text is repeated in the table 3. Is it necessary to give same data in the text and the table?

Moreover, the percent data for Oxford-AstraZeneca vaccine differs in the text and the table 3. (349 reports (49·5%) vs 349 (17.6%). Please correct.

Page 6. lines 226-231

The last paragraph must be deleted.

Do you really think that the risk of COVID-19 related thrombotic events are minimal and likely manageable with available treatments??

Maybe you mean that risk of vaccine related thrombotic events?

Author Response

(The authors gave the same response as above.)

Reviewer 3 Report

In this manuscript by Tobaiqy et al., the authors describe a statistical analysis of thrombotic adverse events reported in the EudraVigilance database for three COVID-19 vaccines (Moderna, Pfizer, and Oxford-AstraZeneca). The authors apply retrospective descriptive statistics to determine characteristics of the population of individuals who developed thrombotic adverse events, particularly pulmonary embolisms, following immunization with one of the COVID-19 vaccines. Given the current COVID-19 pandemic and the global widespread use of Moderna, Pfizer, and Oxford-AstraZeneca vaccines, studies that specifically focus on serious post-vaccination adverse events are needed to inform and encourage clinicians to be vigilant in monitoring patients for developing signs and symptoms of thrombotic adverse events. The research described in this study is timely and needed. Overall, the manuscript is well-written; the reported research is significant and should be of interest to a broad audience.  My only comment is for the authors to correct the misspelling on line 74 ("scare" should be "scarce").  I recommend publishing the manuscript in its present form.       

Author Response

Reviewer#3 correct typo line 74 – scare should be scarce – amended, thank you for noticing, much appreciated